# Study of the Influencing Factors on the Small-Quantity Fuel Injection of Piezoelectric Injector

**DOI:** 10.3390/mi13050813

**Published:** 2022-05-23

**Authors:** Zhenming Liu, Nan Liu, Jingbin Liu

**Affiliations:** 1College of Power Engineering, Naval University of Engineering, Wuhan 430033, China; liuzhenmingyk@163.com (Z.L.); bin_lj@126.com (J.L.); 2Electro-Mechanical Department, Naval Petty Officer Academy, Bengbu 233012, China

**Keywords:** fuel injection, small-quantity, piezoelectric injector, structural parameter

## Abstract

The piezoelectric injection-system provides a reliable approach for precise small-quantity fuel injection due to its fast, dynamic response. Considering the nonlinearity of a piezoelectric actuator, the complete electro-mechanical-hydraulic model of the piezoelectric injector was established and verified experimentally, which showed that it could accurately predict the fuel injection quantity. The small-quantity fuel injection with different driving voltages, pulse widths, and rail pressures was analyzed. The effects of key structural parameters of the injector on the delivery, control-chamber pressure fluctuation, and small-quantity injection characteristics were studied. The results show that the linearity of the curve of the injection volume with the pulse width was relatively poor, and there was a significant inflection point when the piezoelectric injector worked in the small pulse width region (PW < 0.6 ms). The bypass valve significantly accelerated the establishment of the control-chamber pressure, reduced the pressure fluctuation in the chamber, shortened the closing delay and duration of the needle valve, and reduced the rate of the fuel-quantity change so that it provided a greater control margin for the pulse width over the same fuel volume change interval. Under the condition of a small-quantity fuel injection of 20 mm^3^, decreasing the inlet orifice diameter and increasing the outlet orifice diameter shortened the minimum control pulse width and fuel injection duration required for the injector injection, which is beneficial for multiple and small-quantity fuel injection. However, these behaviors reduce the control margin for the pulse width, especially in small pulse width regions.

## 1. Introduction

The high-pressure common rail injection-system has the characteristics of a high injection pressure and an adjustable injection rate. The system can independently adjust the fuel injection parameters without being affected by the engine operating conditions and can effectively reduce the emissions and fuel consumption of a diesel engine, making it the mainstream of current diesel fuel injection systems. As the core component of the fuel-injection system, most injectors currently use solenoid valves as the drive control mechanism. This approach requires a higher energy loss when the injector is working [1]. When the common rail fuel pressure is high, the injector has an obvious fuel leakage, resulting in a waste of energy [2]. In addition, the gap between the electromagnetic coil and the armature is the largest when the solenoid valve is opened. The electromagnetic force to drive the ball valve is minimal, increasing with the square of the electric current. This process wastes 80–100 μs before the current is large enough to generate a sufficient electromagnetic force to open the ball valve, which is not conducive to the rapid response of the injector [3].

One effective method is to use a piezoelectric actuator as the mover of the injector. Piezoelectric injection technology, based on the inverse piezoelectric effect of piezoelectric materials [4], and is an emerging drive control technology, has the advantages of a large driving force, fast response, and low-power consumption compared to electromagnetic injectors. In particular, its faster dynamic response improves the stability and reliability of the injection system and enables a more flexible and high-pressure fuel injection [5]. In 2016, the new piezoelectric common rail injection system, developed by Continental AG, achieved a high rail pressure of 250 MPa and a multiple injection of up to 8 times per working cycle with short injection intervals of 150 μs [6].

The precise control of small-quantity fuel injection is one of the main technical characteristics of high-pressure common rail fuel systems. The small-quantity and multiple injection processes can effectively improve the fuel spray process and reduce the impinging fuel volume in the cylinder [7]. In addition, the disturbance effect of fuel injection can be fully utilized by adjusting the injection volume and injection interval, thus increasing the mixing rate of the fuel and gas, improving the combustion process in the cylinder and reducing the emissions of soot and NOx [8]. The control characteristics of the high-pressure common rail fuel injection system on small-quantity fuel injection largely determine its control precision and stability for the multiple-injection fuel quantity. At the same time, the complex structural parameters of the piezoelectric injector are directly related to the internal fuel flow and needle response, which in turn affects the fuel injection process of the injector [9,10]. Based on the above phenomenon, the author conducted a simulation study on the response characteristics and small-quantity characteristics of an indirect-drive piezoelectric injector with a bypass valve under different control strategies and structural parameters.

However, compared to the solenoid valve injectors, the difficulty in simulating piezoelectric injectors is how to create mathematical models of a piezoelectric actuator. Some studies have been conducted by previous researchers. For example, Jinwook Lee [11] established a linear mathematical model of a piezoelectric actuator based on the classical piezoelectric equation, and then established a mathematical model of the piezoelectric injector using AMESim to analyze the needle valve response and injection rate. Chris A. Satkoski [12] and Poulyaev [13] established a mathematical model of the injector that included a piezoelectric actuator linear model to calculate the injection time and rate. In the application process, the above models either have low-simulation accuracy, difficult parameter identification, or require a large number of test data for support. Although there are many static, high-precision, nonlinear mathematical models regarding actuators, they are not suitable for the high-frequency dynamic responses of piezoelectric injectors.

In this paper, a nonlinear mathematical model of a piezoelectric actuator was established using MATLAB/Simulink to describe the electro-mechanical conversion process of the actuator, and a nonlinear mathematical expression of the actuator output displacement was achieved. Then, a complete one-dimensional electro-mechanical-hydraulic model of the piezoelectric injector was established through the joint simulation of Simulink and AMESim. The response characteristics and small-quantity characteristics of the piezoelectric injector under different rail pressures, driving voltages, and control pulse widths were systematically analyzed. The effects of the bypass valve, control chamber inlet and outlet diameter on the pressure fluctuation, and the small-quantity characteristics of the injector chamber during the injection process were studied to improve the small-quantity injection process.

## 2. Piezoelectric Injector Working Principle

The piezoelectric injector studied in this paper is shown in Figure 1. The injector achieves movement of the ball valve through the actuator constituted by the piezoelectric laminated stack. When the injector does not work, the cone valve is closed completely due to the effect of resetting spring 8 and the fuel in the chamber. The bypass valve (plane valve) is open completely. The needle is close to the needle valve seat under the joint action of the high-pressure fuel in the control chamber and the needle valve spring, thus keeping the needle valve closed. When the piezoelectric actuator is electrified, it stretches downward due to the inverse piezoelectric effect, and amplifies the displacement by the displacement amplifier to make the ball valve open. The bypass valve closes, and the fuel in the control chamber leaks through the outlet, which makes the pressure in the control chamber drop. The needle moves upward under the action of the delivery-chamber and the sac-chamber, and the injection starts. When the voltage on the piezoelectric actuator is unloaded, the actuator retracts upward, the ball valve closes, and the bypass valve opens. The fuel flows into the control chamber rapidly through the intakeII and the bypass valve, which makes the pressure in the control chamber rise, and the needle quickly settle down, ending the fuel injection.

## 3. Piezoelectric Injector Model

### 3.1. Mathematical Sub-Model of Piezoelectric Actuators

The key to the simulation of the piezoelectric injector lies in the mathematical expression of the piezoelectric actuator inside the injector. There are models based on the mathematical description of the hysteresis mechanism such as the Preisach model [14], Bouc-Wen model [15], Maxwell sliding model [16], etc. However, these are not suitable for the high-frequency dynamic response of piezoelectric injectors.

In fact, the nonlinearity of the piezoelectric actuators is caused by the domain rotation and saturation of piezoelectric ceramics [17]. In this paper, the classical piezoelectric equation was used to linearly describe the electro-mechanical conversion process of the actuator. In addition, considering the coupling force field between the reset spring, the hydraulic amplifier, and the actuator, the ferroelectric domain of the piezoelectric material is turned over, which results in a change in the polarization intensity of the actuator stack and a change in the electric field inside the piezoelectric stack. Finally, the output displacement of the actuator is affected. A nonlinear mathematical model of a piezoelectric actuator was established based on MATLAB/Simulink to describe the electro-mechanical conversion process of the actuator. Additionally, the nonlinear change in the internal electric field caused by the stress change on the electrode surface of the actuator was taken into account. The nonlinear expression of the displacement of the actuator is then presented.

The piezoelectric actuator is a crystal stack composed of n-layers of piezoelectric ceramic sheets with a thickness d. The structure is shown in Figure 2. The area of each layer of the ceramic sheet is l×w. The piezoelectric actuator is mechanically connected in series and the circuit is connected in parallel.

When the piezo-actuator is electrified, the following is true, according to the classical piezoelectric equation:(1)$$\left\{\begin{array}{l} D_{3}=\varepsilon_{33}E_{3}+d_{33}T_{3}\\ S_{3}=d_{33}E_{3}+\frac{1}{Y_{33}}T_{3} \end{array}\right.$$
where the first character in the subscript of the variable represents the direction of the electric field, and the second represents the direction of the displacement of the actuator. *D*_3_ = *q*/*A*, where q is the total charge of the piezoelectric actuator, which can be obtained from Equation (2) as follows:(2)$$q=nA\varepsilon_{33}E_{3}+nAd_{33}T_{3}=nA\varepsilon_{33}\frac{v}{d}+nAd_{33}T$$

Piezoelectric crystals are anisotropic, and their polarization vectors are not in the same direction as the electric field. When the polarization direction is the same as the expansion direction, the dielectric constant under mechanical clamping is as follows:(3)$$\varepsilon_{33}^{}=Cd/A$$

Therefore, the total charge of the driver is as follows:(4)$$q=nCv+nAd_{33}T_{3}$$

The strain *S*_3_ was obtained through Equations (1) and (4) as follows:(5)$$S_{3}=\frac{1}{nA}\frac{d_{33}}{\varepsilon_{33}}q+(1-k^{2})\frac{1}{Y_{33}}T_{3}$$

After transformation of the formula as follows:(6)$$T_{3}=\frac{Y_{33}}{1-k^{2}}(S_{3}-\frac{1}{nA}\frac{d_{33}}{\varepsilon_{33}}q)$$
where *k*_2_ is the electromechanical coupling coefficient and is calculated by the following formula:(7)$$k^{2}=Y_{33}d_{33}^{2}/\varepsilon_{33}$$

The strain *S*_3_ calculated from the displacement of the piezoelectric actuator is as follows:(8)$$S_{3}=K_{1}x$$
where *K*_1_ = 1/*nd*. The force Fa gained by stress is as follows:(9)$$F_{a}=AT_{3}$$

The stress causes the displacement of the electric potential to change the polarization *P* of the piezo-electric material. According to the Ising spin model, the nonlinear characteristic between the polarization intensity *P* and polarization electric field intensity *E* can be expressed by the following expression [18]:(10)$$P=P_{s}\tanh\left(\frac{E_{3}}{a}\right)$$
where *P*_s_ is the saturated polarization strength of the piezoelectric actuator material that is obtained from the P-E test curve of the ceramic material (shown in Figure 3). The temperature of the model is not considered yet, and the P-E test curve at 30 °C is used. Therefore, the intensity of the electric field is as follows:(11)$$E_{3}=a\tanh^{-1}\left(\frac{P}{P_{s}}\right)$$

Assuming that the polarization intensity does not exceed the saturated polarization intensity, the field strength *E*_3_ is expanded by a power series, and the nonlinear relationship is expressed as follows:(12)$$E_{3}=a\left[\left(\frac{P}{P_{s}}\right)+\frac{1}{3}{\left(\frac{P}{P_{s}}\right)}^{3}+\frac{1}{5}{\left(\frac{P}{P_{s}}\right)}^{5}\right]$$

The nonlinear characteristics of the polarization and electric field is shown in Figure 4.

Based on the above model, a nonlinear dynamic model of the piezoelectric actuator was established, as shown in Figure 5.

### 3.2. Models of Mechanical Hydraulic Components

The AMESim software was used to simulate the other mechanical-hydraulic system components of the injector, as shown in Figure 6.

The model simulates each part of the injector through the mechanical and hydraulic components. The input parameters of each component calculation module were set according to the actual structural parameters and operation parameters of the injector. The calculation equations of the internal functions of the components include:

The continuity equation of the fluid flow is as follows:(13)$$\int \rho udA_{ori}=0$$
where *ρ* is the fuel density; u is the fuel flow rate; and *A*_ori_ is the effective flow area of the hole.

The compressibility equation of the fluid is as follows:(14)$$dP_{f}=-\frac{E}{V}dV$$
where *P_f_* is the fuel pressure in the chamber, *E* is the fuel elastic modulus, and *V* is the chamber volume.

The fuel flow in and out of the chamber is as follows:(15)$$Q=\mu A_{ori}\sqrt{\frac{2\Delta P}{\rho }}$$
where *μ* is the flow coefficient of the hole; ΔP is the pressure difference on both sides of the hole.

### 3.3. Model Validation

#### 3.3.1. Model Validation of the Piezoelectric Actuator

The output displacement of the piezoelectric actuator is tested on the dynamic characteristic test bench, and the experimental results are compared with the numerical results. The basic parameters of piezoelectric actuators are shown in the Table 1. The experimental system diagram is shown in Figure 7. Force regulation is achieved by installing spring sheets with the same stiffness and different thickness in the fixture loaded with piezoelectric actuators. Temperature regulation is achieved by placing the piezoelectric actuators in a constant temperature drying box.

A self-developed driving circuit device was used to change the driving voltage by adjusting the DC power supply voltage and the driving current by adjusting the current limiting resistance. In addition, the driver circuit can adjust the pulse width, interval, and working frequency of multi-pulse (1–3 times) control signals. A laser Doppler micrometer was used to detect the displacement signal, which included an OFV-2510 vibration test controller, an OFV-534 compact optical head, and an external laser probe. It adopts the principle of fringe counting and is suitable for vibration measurements with a certain amplitude. The maximum test frequency was 250 kHz. The test data were recorded by a Tektronix TDS2022B oscilloscope that had a two-channel band width of 200 MHz and a sampling rate of 2 GS/s.

The dynamic test results of the actuator displacement were compared with the numerical results, as shown in Figure 8. The driving voltage was 140 V. The control pulse width of the two driving processes was 0.55 ms and 2.2 ms, respectively, and the control signal interval was 1.2 ms. From the comparison results, the simulation value of the output displacement was in good agreement with the test value, so the model could better reflect the dynamic characteristics of the actuator.

#### 3.3.2. Model Validation of the Injector

The injection characteristics of the piezoelectric injector were tested by an EMI 2 instantaneous injection law tester on the common rail system injection characteristics test platform. The test platform is shown in Figure 9. Figure 10 shows the comparison between the experimental and numerical results of the injection rate at a rail pressure of 1200 bar and an injection pulse width of 1.5 ms. Figure 11 shows the comparison between the experimental and numerical results of the injection rate at a rail pressure of 900 bar and an injection pulse width of 1 ms. It can be seen that the simulation value and the test value of the injection rate curve were basically fitted, and the model met the calculation accuracy requirements.

## 4. Injection Characteristics Analysis of the Piezoelectric Injector

### 4.1. Analysis of the Fuel Quantity Characteristics of the Injector

Figure 11 shows the effect of the injection pulse width on the injection characteristics under different common rail pressures. From the characteristics of the fuel injection, it can be seen that the relationship between the injection volume and control pulse width was linear in the large pulse width region. In the small pulse width region (0.1–0.6 ms), the linearity of the injection volume with the pulse width was poor, and the rate of change increased with an increase in the pulse width. The curve had an obvious inflection point. When the common rail pressure was 1000 bar, the injector began to work after the pulse width reached 0.2 ms. With an increase in the common rail pressure, the change rate in the fuel volume with the pulse width increased, and the curve inflection point moved to the small pulse width area.

The injection pressure increases with an increase in the injection pulse width; the higher the common rail pressure, the greater the change rate in the injection pressure. When the injection pulse width reaches an inflection point, the injection pressure remains unchanged. With an increase in the common rail pressure, the curve inflection point moves to the small pulse width area. In addition, compared to a rail pressure of 1000 bar, the pressure loss in the injection pressure (the difference between the rail pressure and injection pressure) clearly increased when the rail pressure was 1800 bar.

The linearity of fuel injection duration Δ*t* varies in the large and small pulse width regions. With an increase in the rail pressure, the change rate in the fuel injection duration in the small pulse width region increases, while that in the large pulse width region decreases. Figure 12 shows the characteristics of the needle movement under a different pulse width. Under the same rail pressure, the opening delay and closing speed of the needle valve remain unchanged with an increase in the pulse width. However, in the small pulse width region, due to the needle valve lift not reaching its maximum, the increase in pulse width extended the duration of the needle valve opening and closing, resulting in a significant change in the injection duration (Figure 12a). In the large pulse width region, after the needle valve lift reached its maximum, the increase in the pulse width only led to an increase in the needle valve’s maintenance time at the maximum lift, causing the variation range of the fuel injection duration to decrease, which is the reason for the inflection point of the fuel characteristic curve.

In addition, with the increase in the rail pressure, the opening speed of the needle valve increases, and the time required to reach the maximum lift decreases, which makes the injection pulse width, which corresponds to the inflection point of the fuel quantity characteristics curve, decrease with the increase in the common rail pressure.

### 4.2. Analysis of the Influence Factors on the Small-Quantity Fuel Injection

#### 4.2.1. Influence of the Control Parameters

The piezoelectric actuator is used to directly control the displacement of the ball valve to change the control-chamber pressure, thereby changing the needle valve lift, which provides a convenient approach to the precise control of the small-quantity fuel injection. The small-quantity characteristics studied in this paper mainly referred to the injection process in the working area of the small pulse width when the needle valve lift did not reach its maximum. To study the influence of the control parameters on the small-quantity characteristics of the injector, a control pulse width of 0.2 ms was selected.

Figure 13 shows the effect of the driving voltage at different rail pressures on the injection volume, injection pressure, and injection delay (from the start of charging to the start of injection).The results showed that when the driving voltage was less than 100 V, the injection pressure and volume increased linearly with an increase in the driving voltage. When the driving voltage was greater than 100 V, the injection pressure increased, and the fuel volume decreased with an increase in the driving voltage. When the rail pressure was at 600 bar, the injector began to work when the driving voltage exceeded 80 V. An increase in the rail pressure led to a decrease in the injection delay. In addition, the injection pressure in the small pulse width region was generally low. When the rail pressure was 1800 bar, the maximum injection pressure only reached 647 bar.

At the same rail pressure, with an increase in the driving voltage, the injection delay gradually decreased. When the driving voltage increased from 80 V to 160 V, the injection delay was shortened by 0.06 ms. This is because the peak value of the driving current increased with an increase in the driving voltage (Figure 14). The response time of the injector decreased due to the increase in the charge and discharge rates.

The critical driving voltage of the injector was 100 V under all rail pressures. This is because when the driving voltage is approximately 100 V, the lift of the ball l_b_ reaches its maximum. Figure 15 shows the influence of the driving voltage on the control-chamber pressure pcc, the delivery-chamber pressure pdc, the lift of the ball valve l_b_, and the lift of the needle valve ln. When the rail pressure was 1800 bar and the driving voltage was less than 100 V, the peak value of the ball valve displacement increased with the driving voltage, indicating an increase in the outlet flow area, which made the control-chamber pressure drop and the needle rise more rapidly. Therefore, the injection pressure and the injection quantity increased obviously. When the driving voltage exceeded 100 V, the lift of the ball valve reached a maximum value of 59 μm. After that, with an increase in the driving voltage, the maximum displacement of the ball valve remained unchanged, and the opening speed and closing delay increased so that the time to maintain the maximum displacement increased. The pressure relief speed of the control chamber increased, and the opening speed and the maximum lift of the needle valve increased. Therefore, the injection pressure and fuel volume increased. However, as the increment of the opening speed and the closing delay of the ball valve decreased with an increasing voltage, the increment of the injection volume also began to decrease (Figure 14) after the voltage exceeded 100 V.

#### 4.2.2. Influence of the Structure Parameters

Due to the fast response of the piezo-actuator, the needle valve closed quickly before reaching the maximum lift, which is beneficial in achieving control of the small-quantity fuel injection. The opening and closing response of the needle valve is mainly affected by the establishment and release of the control-chamber pressure, which is related to the passage structure parameters. Therefore, the influence of the bypass orifice diameter of the ball valve chamber and the diameter of the control chamber inlet and outlet orifice on the small-quantity fuel injection was investigated in this study.

(1)Influence of the bypass valve

To improve the response speed of the injector, the injector was equipped with a bypass valve from the delivery chamber to the ball valve chamber, so that the control-chamber pressure rapidly increased during the closing phase of the needle valve. Figure 16 reports the fuel quantity characteristics in the closed and open state of the bypass valve when the rail pressure was 1800 bar and the driving voltage was 100 V. The bypass valve significantly accelerated the establishment of the control-chamber pressure, reduced the pressure fluctuation in the chamber, shortens the closing delay and duration of the needle valve, and reduced the rate of the fuel-quantity change so that it provided a greater control margin for the pulse width over the same fuel volume change interval. At the same time, the injection pressure was reduced by an average of approximately 140 bar in the small pulse width region (0.1~0.6 ms) and approximately 50 bar in the large pulse width region (0.6~1.4 ms).

In addition, the bypass valve had less of an effect on the needle valve opening delay and opening speed, but changed the maximum lift, closing delay, and closing duration of the needle valve. Figure 17 reports the displacement of the needle valve ln and the pressure fluctuations of the control-chamber pcc and delivery-chamber pdc in the closed and open state of the bypass valve. Comparing the dynamic characteristics of the needle valve in the two states, it was determined that the maximum lift of the needle valve of the improved injector was reduced by 27 μm, and the closing time was shortened by 0.058 ms, which is beneficial to the quick stop of the fuel injection.

These findings are attributed to the fact that at 0.72 ms, the high-pressure fuel in the delivery-chamber flows into the ball-valve chamber through the bypass valve, and then flows into the control-chamber through the outlet orifice, so that the control-chamber pressure rapidly rises, and the pressure curves under the two conditions diverged after 0.72 ms. A decrease in the delivery-chamber pressure and an increase in the control-chamber pressure together results in an increase in the hydraulic pressure that drives the needle valve downward, which accelerates the speed of the downward needle movement.

(2)Influence of the inlet and outlet orifices

During the fuel flow process, the inlet and outlet orifices are directly related to the change in the control-chamber pressure, thereby changing the hydraulic pressure acting on the upper end of the needle valve, which ultimately affects the moving response of the needle valve. Due to the limitation of the injector size, the length of the inlet and outlet fuel passages is short, and the main factor affecting the fuel flow capacity is the diameter of the orifices. The effect of both orifices on the fuel injection is interactive. To analyze the influence of the inlet–orifice diameter (IOD) and outlet–orifice diameter (OOD) on the small-quantity fuel injection, under the condition that the IOD is smaller than the OOD and the control-chamber fuel can be smoothly drained, the author selected the structural parameters of the IOD at 0.18~0.24 mm and the OOD at 0.28~0.36 mm.

Figure 18 and Figure 19 show the minimum pulse width and injection duration required for a small-quantity fuel injection of 20 mm^3^ at a common rail pressure of 1800 bar and a driving voltage of 100 V for each inlet and outlet orifice configuration parameter. Decreasing the IOD and increasing the OOD can reduce the minimum control pulse width as well as the fuel injection duration, which is conducive to achieving small-quantity fuel injection. The sensitivity of the control pulse width and injection duration to the variation of IOD and OOD decreased gradually, especially the injection duration, in the regions of ①~③ and ⑥~⑧ in the diagram. Therefore, the choice of orifice diameters in the areas of ③, ④, ⑧, and ⑨ should be considered, as they can effectively reduce the control pulse width and injection duration and avoid the loss of the fuel injection energy caused by excessive outlet orifice diameters.

Figure 20 shows the injection rate, injection pressure, and needle valve lift of the injector with different inlet and outlet diameters when the injection volume was 20 mm^3^. It can be seen that an increase in the OOD led to decreases in the needle valve opening delay, opening duration, and closing delay. The entire injection process moved forward. At the same time, the injection pressure increased, indicating that the injection performance had been improved. A decrease in the IOD also caused the entire injection process to move forward. The injection duration was slightly shortened, and the maximum injection pressure dropped slightly.

Therefore, in order to improve the response characteristics of the needle valve opening process, the OOD to the IOD can be increased appropriately. However, an excessive increase will slow down the establishment of the control-chamber pressure and delay the closing duration of the needle valve, which is not conducive to the quick stop of the fuel injection.

Figure 21 shows the variation in the fuel injection volume with the control pulse width under three conditions: the reference structure parameter (IOD = 0.24 mm, OOD = 0.28 mm), the IOD reduced to 0.2 mm, and the OOD increased to 0.32 when the rail pressure was 1800 bar. It can be seen that reducing the IOD and increasing the OOD increased the change rate of the fuel injection rate in the small pulse width region, which reduced the control margin of the fuel injection, but increased the fuel injection rate. This is conducive to shortening the injection duration, thus achieving multiple and small-quantity fuel injection.

## 5. Conclusions

A linear model of the piezoelectric actuator was established based on the classical piezoelectric equation. On this basis, a nonlinear mathematical model of the actuator was realized by taking the nonlinear change of the internal electric field caused by the change in stress on the electrode surface of the actuator into account. Moreover, a one-dimensional electro-mechanical-hydraulic coupling model of the piezoelectric injector was established that could effectively predict the actuator displacement and injection process.

When the piezoelectric injector worked in the large pulse width region (PW >0.6 ms), the fuel injection volume and the control pulse width were linearly related. However, in the small pulse width region (0.1–0.6 ms), the linearity of the fuel volume with the pulse width was poor, and the change rate increased with the pulse width. The fuel volume curve had an obvious inflection point that moved to the small pulse width region with an increase in the common rail pressure.

Under the condition of a small control pulse width (PW = 0.2 ms), the peak current of the actuator increased with an increase in the driving voltage, the response time of the injector was shortened, and the injection pressure and fuel quantity increased linearly. When the driving voltage was greater than 100 V, the increment of the injection pressure and fuel quantity decreased with an increase in the driving voltage. The critical driving voltage of the injector was approximately 100 V under each rail pressure due to the limitation of the lift of the ball valve and the flow capacity of the outlet orifice.

The bypass valve significantly accelerated the establishment of the control-chamber pressure, reduced the pressure fluctuation in the chamber, shortened the closing delay and duration of the needle valve, and reduced the rate of the fuel-quantity change so that it provided a greater control margin for the pulse width over the same fuel volume change interval.

With a small-quantity fuel injection of 20 mm^3^, decreasing the inlet orifice diameter and increasing the outlet orifice diameter shortens the minimum control pulse width and fuel injection duration required for the injection, which is beneficial for multiple and small-quantity fuel injection. However, these behaviors reduce the control margin for the pulse width, especially in the small pulse width regions, which makes precise control more difficult.

## Figures and Tables

**Figure 1 micromachines-13-00813-f001:**
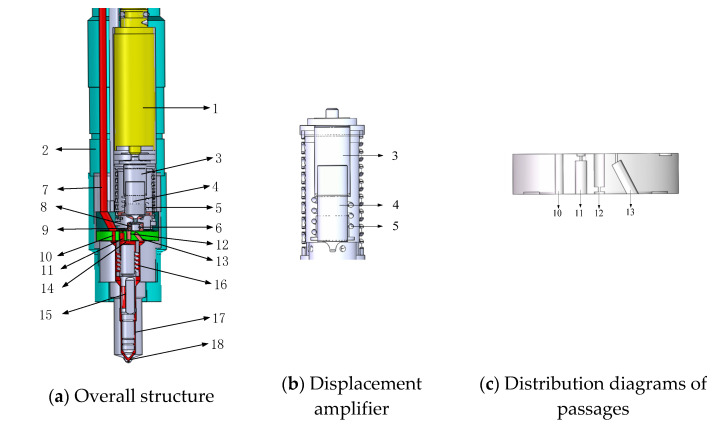
The schematic diagram of the piezoelectric crystal stack.

**Figure 2 micromachines-13-00813-f002:**
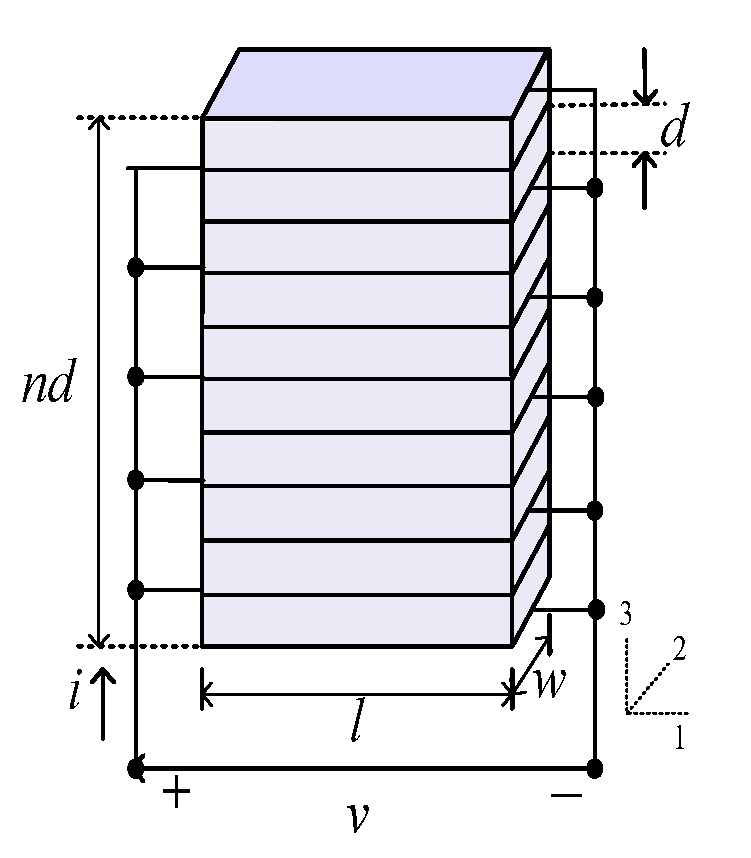
The schematic diagram of the piezoelectric crystal stack.

**Figure 3 micromachines-13-00813-f003:**
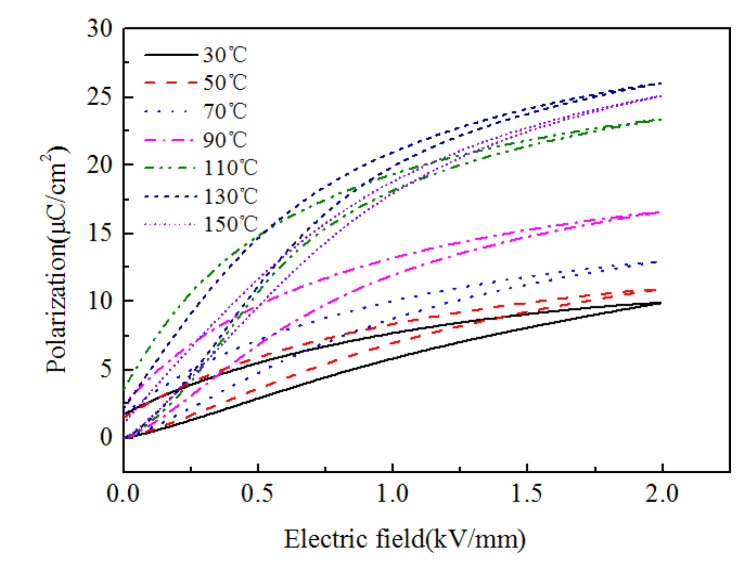
The P-E test curve of the ceramic material.

**Figure 4 micromachines-13-00813-f004:**
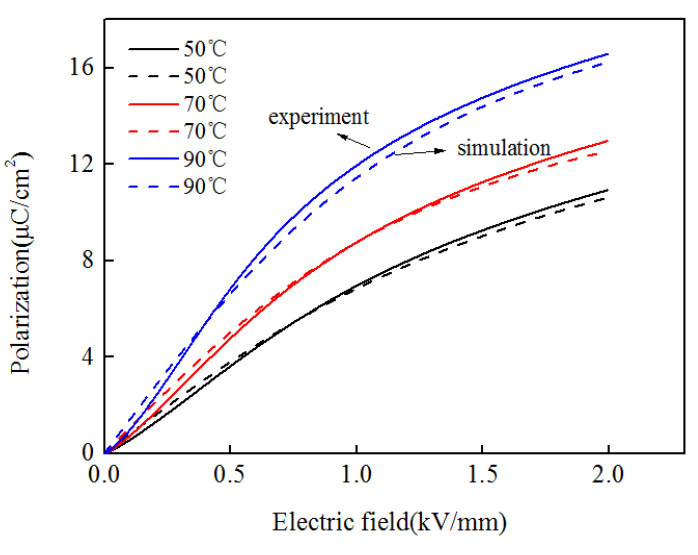
The nonlinear relationship between the material polarization and electric field strength.

**Figure 5 micromachines-13-00813-f005:**
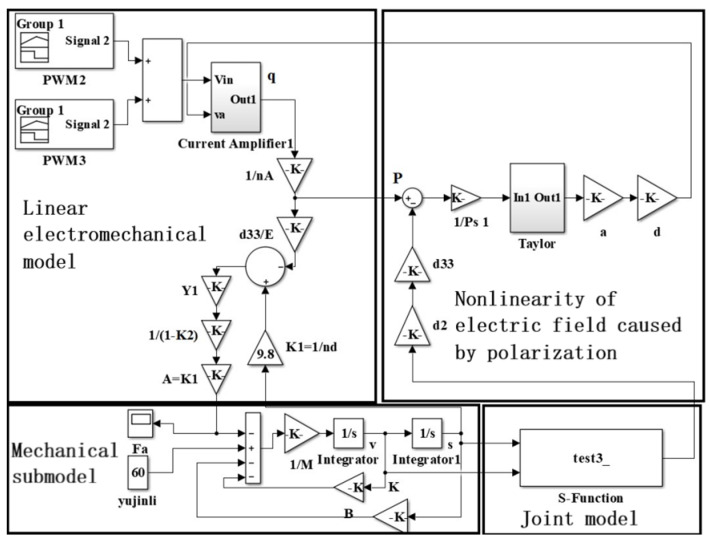
The nonlinear model of the piezoelectric actuator.

**Figure 6 micromachines-13-00813-f006:**
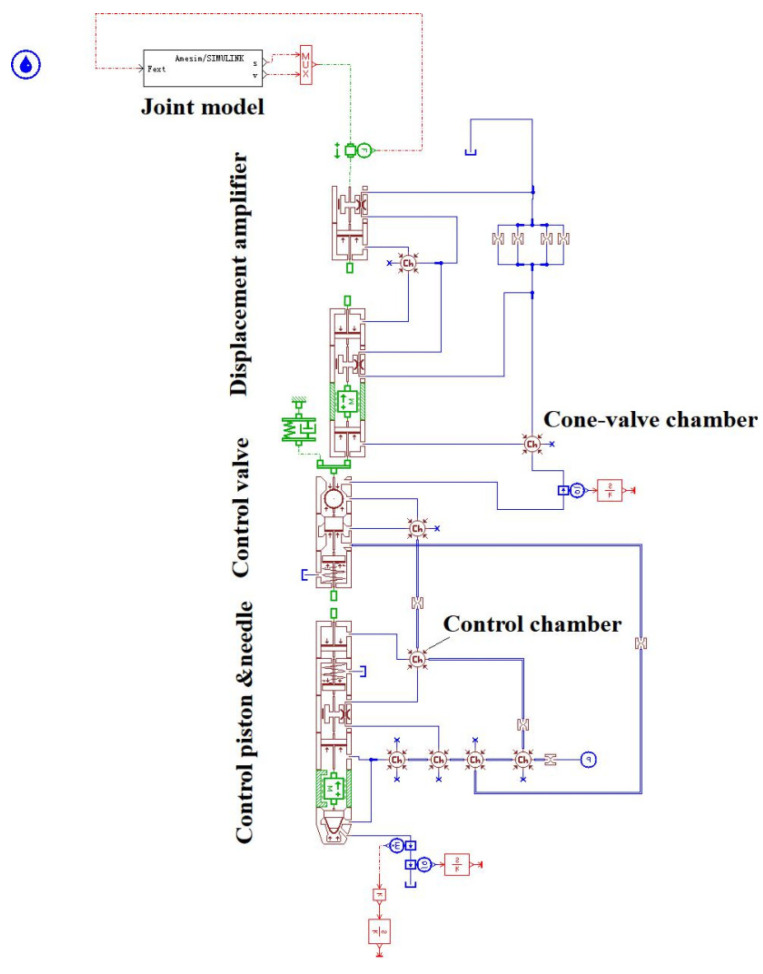
The mechanical-hydraulic model of the injector.

**Figure 7 micromachines-13-00813-f007:**
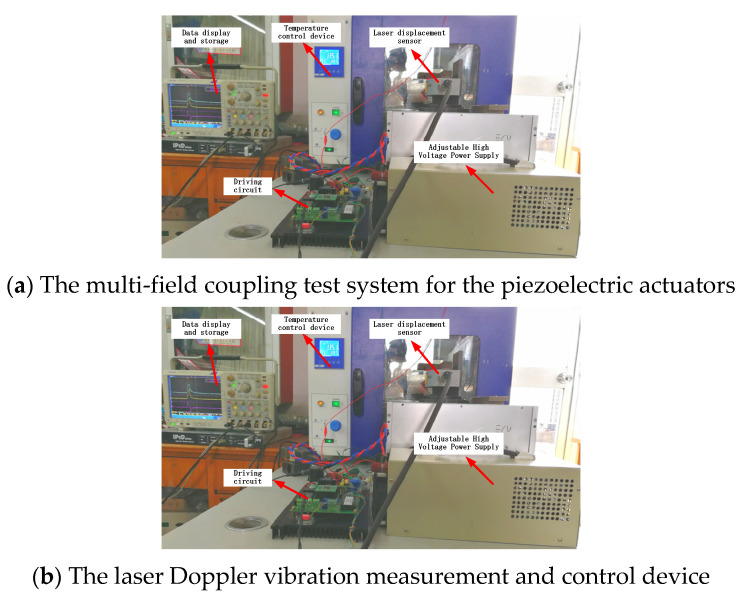
The experimental system equipment.

**Figure 8 micromachines-13-00813-f008:**
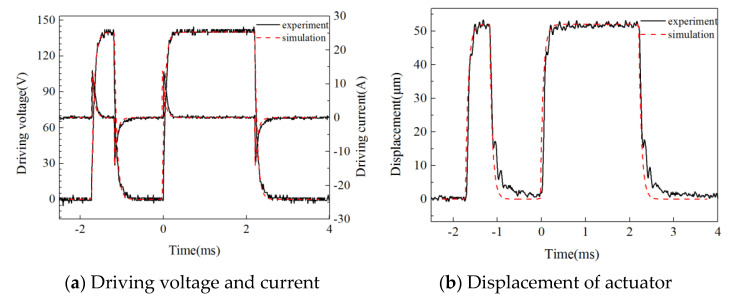
The model validation of the actuator.

**Figure 9 micromachines-13-00813-f009:**
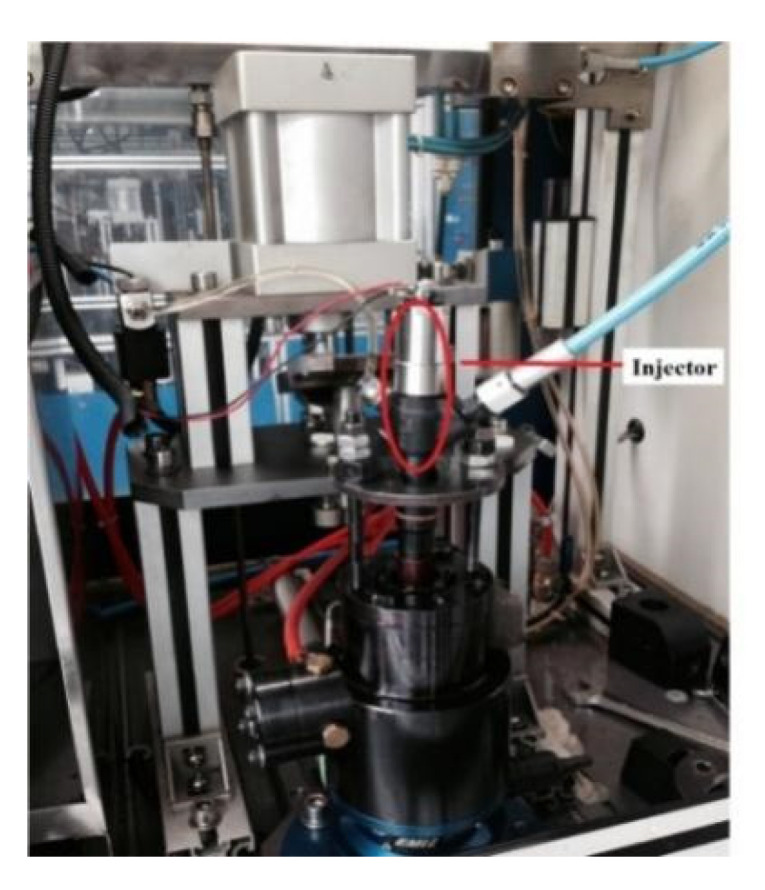
The injection characteristics test platform.

**Figure 10 micromachines-13-00813-f010:**
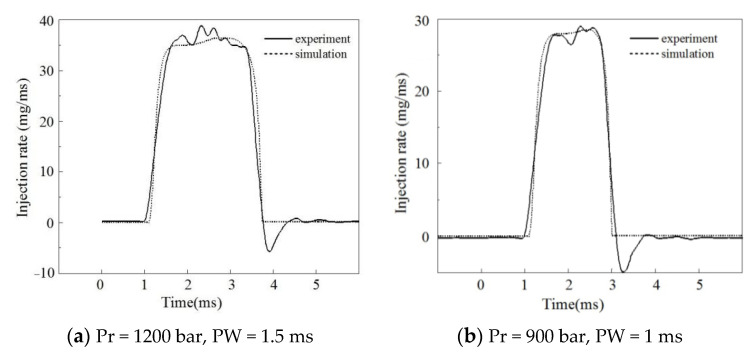
The model validation of the injector.

**Figure 11 micromachines-13-00813-f011:**
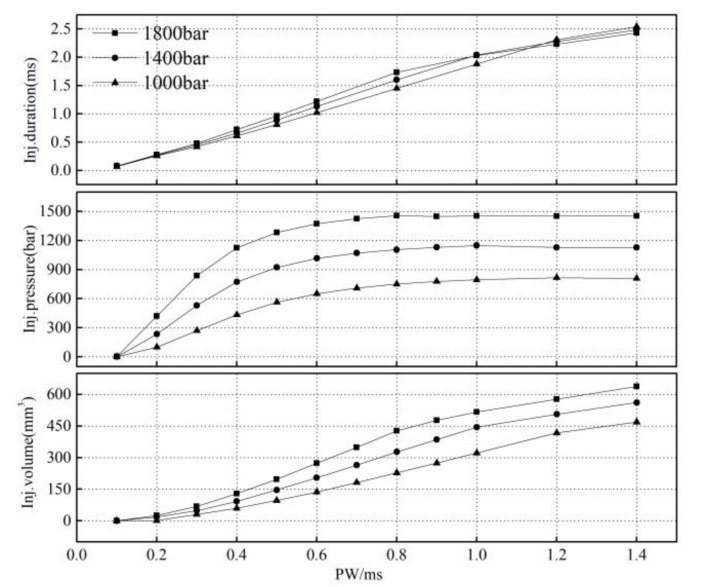
The injection characteristics under different rail pressures and control pulse widths.

**Figure 12 micromachines-13-00813-f012:**
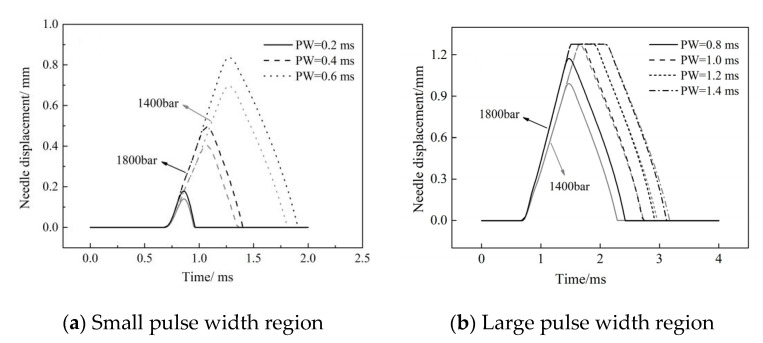
The needle lift curve under different pulse widths.

**Figure 13 micromachines-13-00813-f013:**
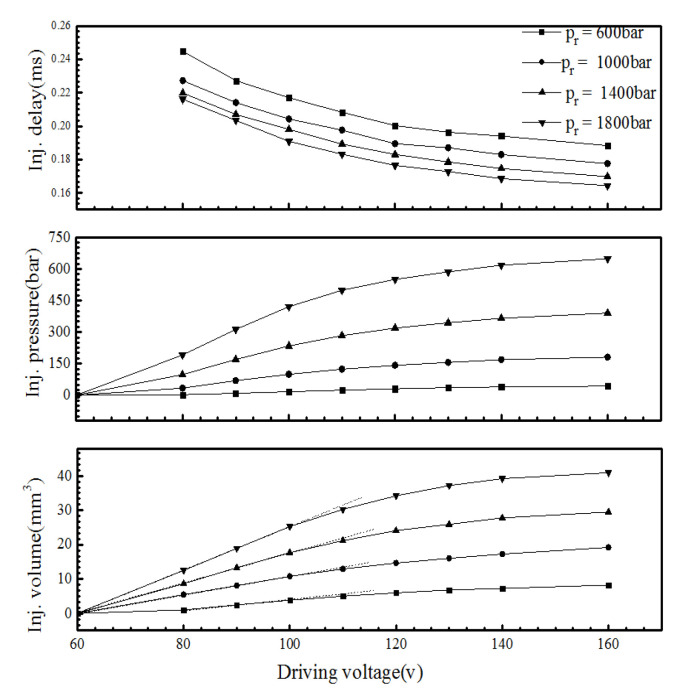
The effect of the driving voltage on the injection volume, injection pressure, and injection delay (PW = 0.2 ms).

**Figure 14 micromachines-13-00813-f014:**
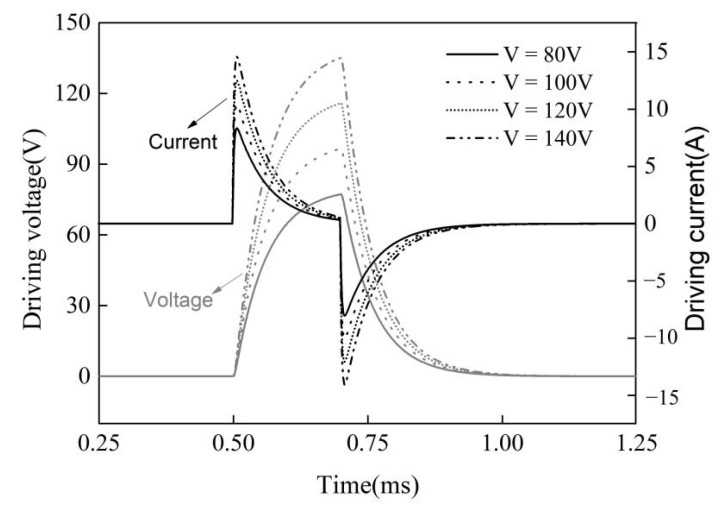
The variation in the driving voltage and current (Pr = 1800 bar, PW = 0.2 ms).

**Figure 15 micromachines-13-00813-f015:**
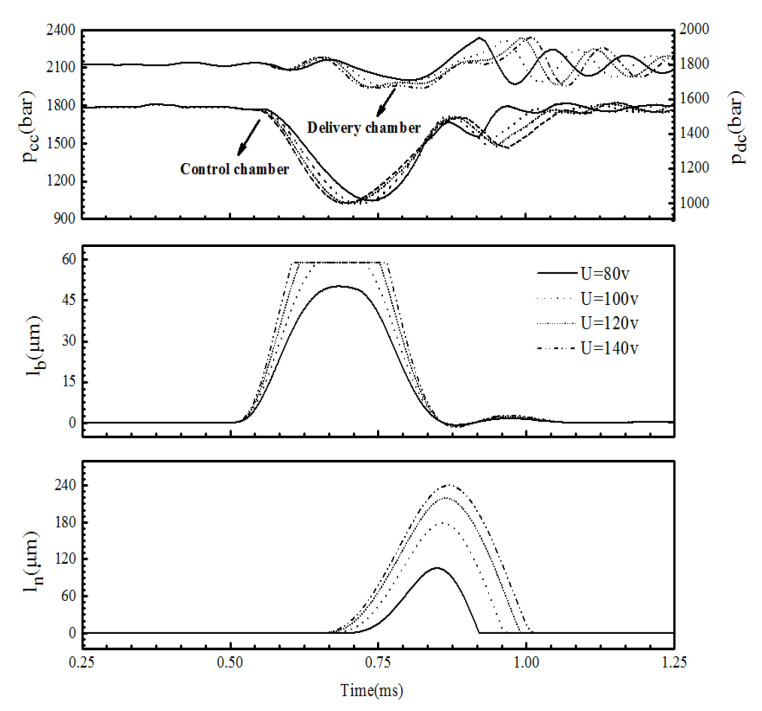
The effect of the driving voltage on the chamber pressure fluctuation and injector response characteristics (Pr = 1800 bar, PW = 0.2 ms).

**Figure 16 micromachines-13-00813-f016:**
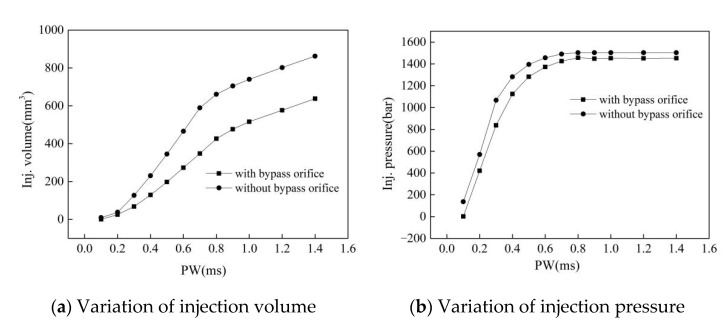
Effect of the bypass valve on the injection volume and pressure (Pr = 1800 bar, U = 100 V).

**Figure 17 micromachines-13-00813-f017:**
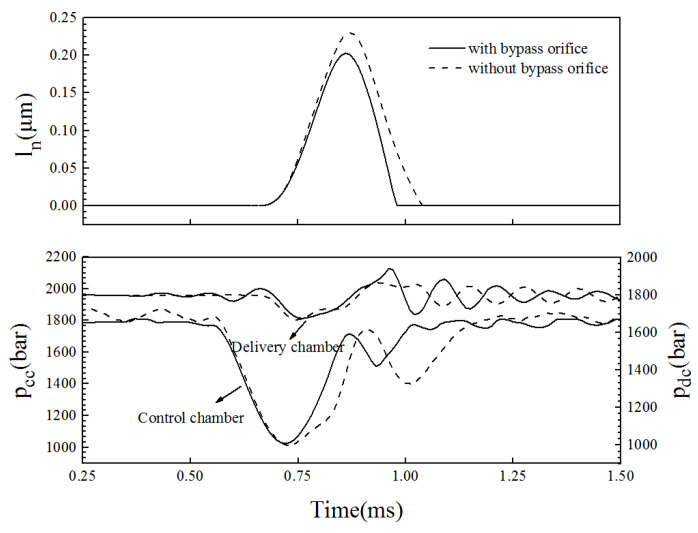
The effect of the bypass valve on the needle response and pressure fluctuation in the chambers (Pr = 1800 bar, U = 100 V, PW = 0.2 ms).

**Figure 18 micromachines-13-00813-f018:**
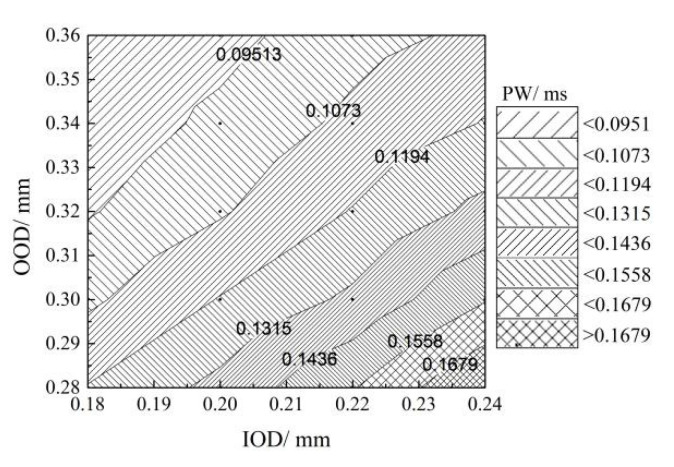
The minimum control pulse width under different inlet and outlet orifice diameters (Pr = 1800 bar, U = 100 V, Q = 20 mm^3^).

**Figure 19 micromachines-13-00813-f019:**
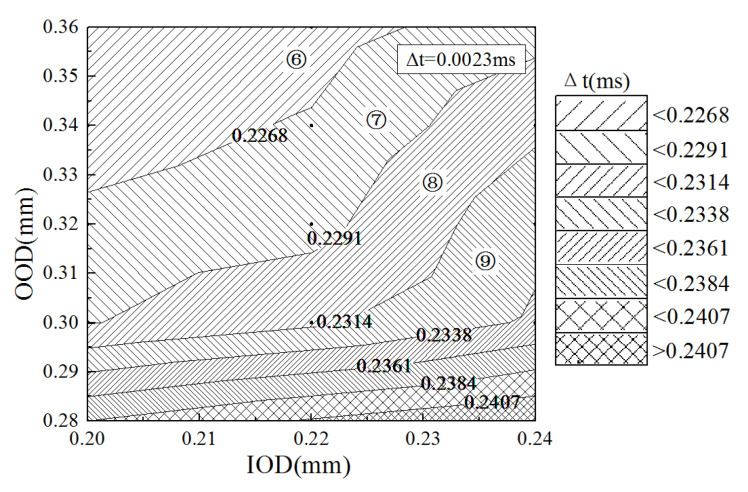
The minimum injection duration under different inlet and outlet orifice diameters (Pr = 1800 bar, U = 100 V, Q = 20 mm^3^).

**Figure 20 micromachines-13-00813-f020:**
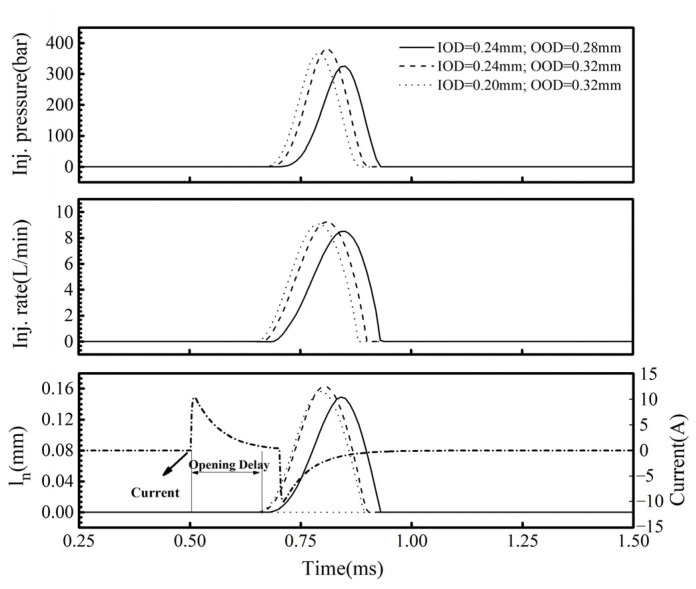
The injection characteristics of the injector at different inlet and outlet diameters under the small-quantity fuel condition of 20 mm^3^ (Pr = 1800 bar, U = 100 V, Q = 20 mm^3^).

**Figure 21 micromachines-13-00813-f021:**
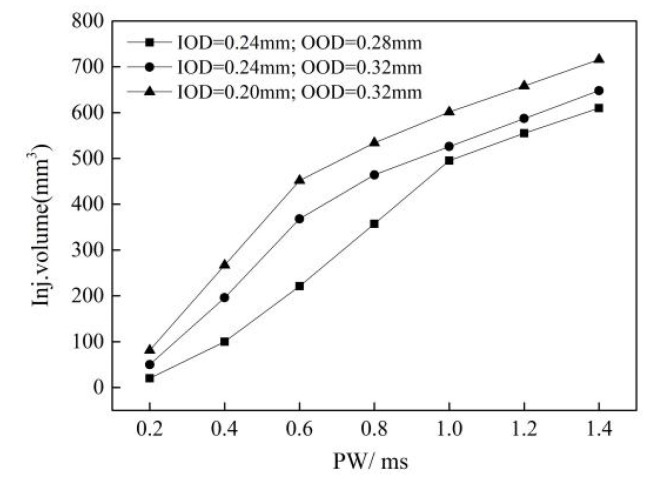
The variation in the fuel injection volume with the control pulse width under different inlet and outlet diameters (Pr = 1800 bar, U = 100 V).

**Table 1 micromachines-13-00813-t001:** The specification of the piezoelectric actuator.

Longitudinal Piezoelectric Constant (m/V)	850
Dielectric constant (F/m)	3.08 × 10^−8^
Elastic modulus (N/m^2^)	4.57 × 10^10^
Layer number of ceramic sheets	600
Ceramic sheet thickness (m)	9 × 10^−5^
Electrode area (m^2^)	25 × 10^−6^
Equivalent capacitance (μF)	5.13
Equivalent impedance (Ω)	0.62
Mass (kg)	0.042
Stiffness coefficient (N/m)	1.5 × 10^7^

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
