# Peer review of "Study of the Influencing Factors on the Small-Quantity Fuel Injection of Piezoelectric Injector"

_micromachines, 2022, doi:10.3390/mi13050813_

Round 1

Reviewer 1 Report

1.      There should be ‘6. cone valve’ in Figure. 1, the serial number 8, 9 in Figure. 1 should also be corrected.
2.      The in-let orifice B is not marked in Figure. 1. If you mean the in-let orifice B is ‘11. in-take â…¡’ in Figure. 1, then it should be unitized in this paper.
3.      Does the inj. pressure in Figure 16 (b) mean the largest injection pressure during the injection?

Reviewer 2 Report

Authors discussed many effects on the small quantity fuel injecion by CFD for a piezoelectric injector. It is interesting and some good conclusion were gotten. It can be accepted with revisions.

  1. why ET represent the pulse width, it is quite strange.
  2. in the parameter, why the driving voltages should be considered? it should be decided originally.
  3. Fig. 5 and 6 are too small to see any details.
  4. the equations (13-15) can be gotten from any textbook, however, I want to know how to decide the u in E15 and C in E3, they are important and have strong effect on your results.
  5. From your results in Fig. 10, CFD are similar to experiment. It shocked me a lot. But how to get the experiment data? which method you used in your test platform? Please explain it.
  6. Besides, in your experiment, what passfilter you used in the processed data to get this injection curves, it is important for your results

Round 2

Reviewer 2 Report

It can be accepted now